# New Genetic Markers Differentiating IPEC and ExPEC Pathotypes—A New Approach to Genome-Wide Analysis Using a New Bioinformatics Tool

**DOI:** 10.3390/ijms24054681

**Published:** 2023-02-28

**Authors:** Marta Majchrzak, Sebastian Sakowski, Jacek Waldmajer, Pawel Parniewski

**Affiliations:** 1Institute of Medical Biology, Polish Academy of Sciences, 106 Lodowa Str., 93-232 Lodz, Poland; 2Faculty of Mathematics and Computer Science, University of Lodz, Banacha 22, 90-238 Lodz, Poland; 3Centre for Data Analysis, Modelling and Computational Sciences, University of Lodz, Scheibler Family Avenue 2, 90-128 Lodz, Poland; 4Institute of Computer Science, University of Opole, Oleska 48, 45-052 Opole, Poland

**Keywords:** genetic markers, genome-wide analysis, data analysis, sequence analysis

## Abstract

The increasingly expanding genomic databases generate the need for new tools for their processing and further use. In the paper, a bioinformatics tool, which is a search engine of microsatellite elements—trinucleotide repeat sequences (TRS) in files of FASTA type—is presented. An innovative approach was applied in the tool, which consists of connecting—within one search engine—both mapping of TRS motifs and extracting sequences that are found between the mapped TRS motifs. Accordingly, we present hereby the tool called TRS-omix, which comprises a new engine for searching information on genomes and enables generation of sets of sequences and their number, providing the basis for making comparisons between genomes. In our paper, we showed one of the possibilities of using the software. Using TRS-omix and other IT tools, we showed that we were able to extract sets of DNA sequences that can be assigned only to the genomes of the extraintestinal pathogenic *Escherichia coli* strains or to the genomes of the intestinal pathogenic *Escherichia coli* strains, as well as providing the basis for differentiation of the genomes/strains belonging to each of these clinically essential pathotypes.

## 1. Introduction

There is a lack of tools for analyzing the enormity of data obtained through NGS sequencing in today’s world. We propose a new search engine based on the use of microsatellite sequences. Microsatellite sequences, also known as Simple Sequence Repeats (SSRs) or Short Tandem Repeats (STRs) [1], which are sequences of between one and six nucleotides in DNA repeated in a tandem fashion are present in genomes of every organism. They are dispersed throughout the genome and can offer the basis for many molecular tools relevant in medical diagnostic procedures, epidemiology, evolutionary examinations, or criminal studies [2,3]. From the point of view of diagnostics and molecular epidemiology of microorganisms, particularly interesting and promising are the trinucleotide repeat sequences that make one of the most numerous groups among microsatellite sequences in bacteria.

Regarding the scope of the abovementioned considerations, a general approach has recently been developed, which has made it possible—with the use of TRS profiling [4] and branching processes—to predict the directions of pathogenicity development in the *E. coli* population [5]. Recent years have seen the appearance of tools that enable the search for microsatellite sequences [6,7]. However, these tools are focused exclusively on determining the place of occurrence of microsatellite sequences on a genome, a particular case of which can be TRS motifs. There is a shortage of informatic tools that simultaneously map the genome using TRS motifs and find DNA sequences between them. Consequently, in this work, the tool called TRS-omix is presented. The tool represents a new engine to search information in genomes and enables both mapping and extracting sequences between the flanking sequences (each flanking sequence includes only one TRS motif).

The analysis of many bacterial genomes shows that the most common microsatellite sequences in genomes represent repetitions of three nucleotides (trinucleotide repeats). Figure 1 shows the distribution of microsatellites in the genomes of *Escherichia coli* UTI89 and *Escherichia coli* O157:H7 Sakai, representing two essential and not always genetically distinguishable pathotypes of this species, ExPEC (Extraintestinal Pathogenic *Escherichia coli*) [8,9,10] and IPEC (Intestinal Pathogenic *Escherichia coli*) [9]. The analysis was performed with MICdb software 3.0 [11] on the MICAS 3.0 platform using *E. coli* UTI89 and *E. coli* O157:H7 strain Sakai genomes available at the database.

The number of all possible trinucleotide sequences (TRS) is 64 (*n^k^* = 4^3^ = 64, all variations of the *k*-elements with repetitions of the set of *n*-elements). For each of these sequences, excluding AAA, CCC, GGG, and TTT, repeating it three times as the co-called TRS motif is considered. The set of all 60 examined TRS motifs arranged in three groups is included in Table 1.

There exist various types of scientific software that can be used for whole-genome analyses of microsatellites’ occurrence in the genome. Of these, there are many that allow searching for tandem repeat sequences or microsatellites, e.g., TRF (Tandem Repeats Finder) [12], IMEx (Imperfect Microsatellite Extractor) [7], and many others. However, to the best of our knowledge, no software has yet been created that provides the ability to extract sequences between such trinucleotide repeats. Importantly, each of the genomes tested will contain a different, more or less similar set of such sequences, depending on the overall similarity of the genomes, making the software potentially useful for comparative genomic analyses. In opposition to other well-known software detecting microsatellites, the TRS-omix search engine presented here finds microsatellites—trinucleotide repeats and also extracts sequences between such trinucleotide motifs. Such an approach can find a number of applications in various aspects of genomics. This work focusses on the bacterial model, specifically the genomes of *Escherichia coli* UTI89 and O157:H7 Sakai. It excludes other examinations, including eukaryotic organisms or other genomes, which is possible with this software.

## 2. Results

### 2.1. A Novel Bioinformatics Tool for DNA and RNA Analysis

The TRS-omix software was developed with the use of the ANSI C language. This makes it possible, on the one hand, to use it on different bioinformatics IT platforms and, on the other to integrate it with libraries of different languages (including: R or Python) which are dedicated to applications of, among others, DNA computing, statistical analysis, or data mining techniques in big data. When the main TRS-omix search engine computational method was constructed, the structures of data which use dynamic memory allocation were designed and applied.

The TRS-omix tool, proposed in this work, includes two types of functionalities. The first one makes it possible to select the examination of genomes with circular structures (e.g., the majority of genomes and bacterial plasmids, mitochondrial DNA and chloroplast DNA) or a linear structure (the majority of eukaryotic genomes). The other functionality enables one to determine the minimal and maximal lengths of sequences searched for, which are found between the flanking sequences (each flanking sequence includes only one TRS motif). Therefore, it is possible to extract DNA sequences of length *k* (*k* is a natural number, where *k* > 0) or of length within a certain range <*a*, *b*> (*a* and *b* are natural numbers, where *a*, *b* > 0 and *a* < *b*).

The schematic diagram of action and information flow in the TRS-omix tool is shown in Figure 2. The input data are the following: file *sequence.fasta* and file *trs.txt*. The first file is one of FASTA type, which contains the examined DNA sequences. The file *trs.txt* is a textual one that contains searched for TRS motifs grouped in classes. In this file, the searched for patterns (TRS motifs) are defined in successive verses, where motifs belonging to one of the twenty possible classes (see Table 1) are inserted within one verse. Each of the verses constitutes a sequence of TRS motifs, while each of the TRS motifs in the given verse is preceded by the “#” sign. An exemplary class that includes the following sequences of nucleotides (TRS motifs): CCGCCGCCG, CGCCGCCGC, and GCCGCCGCC in file *trs.txt* are defined by the following verse: #CCGCCGCCG#CGCCGCCGC#GCCGCCGCC. The output data of the engine of TRS-omix are information on sequences existing between the flanking sequences (each flanking sequence includes only one TRS motif) that are found in the textual file *interiors.txt*.

The software was elaborated in such a manner as to enable searching TRS motifs in FASTA files downloaded, for instance, from GenBank—the file called *sequence.fasta*. After starting the software, the examined linear or circular structure ought to be selected and then the determined value of minimal or maximal length of the sequence searched for. In the case where the minimal value is identical with the maximal one, we want to extract the sequences found between TRS motifs of a concrete length. In the case where these values vary, we define extracting sequences of the existing TRS motifs of the length found within the defined range. On finishing the work of the search engine, the file *interiors.txt* is returned (see Figure 3).

### 2.2. Experimental Validation

The functioning of the software was tested on FASTA type files containing different nucleotide sequences, e.g., *Human chromosome* 14—complete sequence, as well as other sequences of bacterial origin (see Table 2), on a computational server. The TRS-omix search engine was tested in both the Linux and Windows operating system environment. The source code and sample data are freely available for download at https://github.com/TRS-omix/software (accessed on 21 February 2023), distributed under the GNU GPLv3 license. The experimental tests were carried out on the basis of genome datasets coming from the NCBI database (https://www.ncbi.nlm.nih.gov).

### 2.3. A general Approach for Differentiation of Organisms Using TRS-Omix

Two genomes of *Escherichia coli* were analyzed—UTI89 [13], representing the ExPEC pathotype [8], the UPEC sub-pathotype [13], and O157:H7 Sakai [14], representing the IPEC pathotype, the EHEC sub-pathotype [15].

As part of the TRS-omix validation, the general approach was to compare the similarity of the sequences generated by the software in such a way as to eliminate those that are identical/similar in both genomes and leave only those that are unique for each of them. Knowing that TRS-omix only searches for sequences between perfect TRS motifs, additional genomic searches with Vector NTI 11.5 software allowed screening out those sequences that had counterparts in both genomes, but their flanking sequences in one of the genomes were imperfect. Finally, the unique sequences of each genome were verified by BLAST N analysis to leave only those with complementary matches only in the genomes of strains belonging to either the ExPEC or IPEC group of pathogens. These sequences have been proposed as specific markers for these *E. coli* pathogenic groups. The general approach is schematically presented in Figure 4 and is described in detail in the following (data analysis with the use of TRS-omix and other IT tools).

### 2.4. Data Analysis with the Use of TRS-Omix and Other IT Tools

To illustrate the applicability of the TRS-omix software in conjunction with other IT tools, an example analysis of the similarity data of two *E. coli* genomes for sequences between (TRS) *n* ≥ 3 motifs will be presented. In this work, we conducted such analyses for two genomes of closely related strains of the *Escherichia coli species*—uropathogenic UTI89 (Acc. No NC_007946.1) and diarrheal O157:H7 Sakai (Acc. No BA000007.3). Genomic sequences flanked by the TRS motif can be flanked on both sides, on the 3′ direction and on the 5′ direction. It was decided mainly to check whether some of the extracted sequences could be specific for the two genomes studied. In addition, to assess whether it is possible to search for sequences specific for the *E. coli* ExPEC or IPEC pathotypes represented by these genomes. The analysis was carried out in the following steps:

**Stage 1.** Use of TRS-omix on each of the genomes. All fragments containing DNA sequences between TRS motifs for the studied UTI89 and O157:H7 genomes were 2640 and 2777, respectively (raw text data, Appendix A). Interestingly, the algorithm used in the MICdb 3.0 software database [11] showed 2371 and 2491 sequences of this type for these genomes, respectively (see Appendix A). Thus, this algorithm did not find a certain number of TRS repeats, indicating the advantage of our software.

**Stage 2.** Comparison of the similarity of sequences from all studied classes between TRS repeats *n* ≥ 3 occurring in both genomes, using the Vector NTI 11.5 software, CLUSTAL W algorithm. Only sequences of at least 100 bp length were considered for comparison in these studies. Such a comparison allowed one to determine which sequences are identical or significantly similar in both bacterial genomes tested, leaving only those that remained specific for one or the other genome. The similarity of the sequences extracted by TRS-omix from both genomes was compared for each class separately. For example, in the case of sequences between (CCG) *n* ≥ 3 and (TCG) *n* ≥ 3 repeats (Appendix A), the analysis showed the presence of a pair of homologous sequences and seven singletons (see Figure 5). Similar analyses were performed for all sequences tested between different combinations of TRS motifs.

**Stage 3.** Checking the actual uniqueness of the sequence. The sequences pre-qualified as unique (seven singletons in the example above) to one genome may still have their counterpart in another genome, which could have occurred due to nucleotide changes (mutations) in the flanking TRS sequence and failure of TRS-omix to find such a sequence (imperfect TRS) or the presence of a different TRS motif within the sequence, which resulted in assigning it to another class. In addition, some sequences were partially homologous to their counterparts. Such a situation can be seen in the UTI-89-1897 1875 bp fragment, which shares homology with 340 bp of the Sakai sequence. Some other genomic rearrangements, such as internal insertions/deletions, may have occurred (SAK-1679 counterpart sequence in the UTI89 genome), leaving homology at 5′- or 3′-parts of sequences, and therefore were not considered truly unique. (Appendix A). Out of seven sequences initially qualified as unique, only two turned out to have no homologous sequences in the second examined genome. There were 7182 bp and 389 bp sequences in the *E. coli* O157: H7 Sakai genome (SAK-436 and SAK-2393, respectively). (Appendix A).

**Stage 4.** Assess whether the selected DNA sequences, unique for a given pathotype, can be potentially helpful in typing the group of ExPEC and IPEC pathogens. Examination of whether the selected sequences are typical only for the genomes of the ExPEC pathotype, represented by the UTI89 or IPEC genome, represented by the O157: H7 Sakai, was carried out by BLAST N (highly similar sequences, megablast) analysis on the NCBI platform (https://blast.ncbi.nlm.nih.gov/Blast.cgi).

The decisive criterion for choosing a given DNA sequence as typical for the ExPEC pathotype was the confirmed lack of homology (1000 maximum target sequences) in the BLAST N database to the genomic sequences of:*Enterobacteriaceae* other than *Escherichia coli*,*Escherichia* other than *E. coli*,*E. coli* belonging to the IPEC pathotype,non-pathogenic *E. coli*.Bacteriophages.

Similarly, the decisive criterion for choosing a given DNA sequence as typical for the IPEC pathotype was the confirmed lack of homology (1000 maximum target sequences) in the BLAST N database to the genomic sequences of:*Enterobacteriaceae* other than *Escherichia coli*,*Escherichia* other than *E. coli*,*E. coli* belonging to the ExPEC pathotypes,non-pathogenic *E. coli*.Bacteriophages.

In the above example (sequences between CCG *n* ≥ 3 and TCG *n* ≥ 3 repeats), only the 389 bp O157: H7 Sakai genomic sequence showed the only similarity to *E. coli* genomes belonging to the IPEC pathotype. Thus, it can be considered a useful marker in the genetic typing of this group of microorganisms. The other sequence (SAK-436) was excluded as it showed homology to other *Escherichia* species, *Shigella* sp., *Citrobacter* sp., *Klebsiella pneumoniae*, *Enterobacter* sp., and *Salmonella* sp.

Taking into account all the investigated sequence classes analyzed and assumed exclusion criteria, 111 sequences from the O:157:H7 Sakai genome and 25 sequences from the UTI89 genome sequences were selected for analysis in terms of their specificity in relation to the ExPEC pathotype or the IPEC pathotype (Appendix A). Five sequences typical for IPEC pathotype strain genomes were found (Appendix A) and two sequences specific for ExPEC genomes (Appendix A). We excluded the sequences UTI-98-1111, UTI-98-1112, UTI-98-1113, UTI-98-1114, UTI-98-1115, and UTI-89-1116 (Appendix A) from potential candidates because they showed homology to the *E. coli* O18ac:H14 genome. This is probably representative of the NMEC [16] sub-pathotype, but we did not find sufficient confirmation for this particular H serotype. These results are summarized in Table 3. The following IPEC genomes, based on unique *E. coli* O:157:H7 str Sakai sequence fragments, were identified by the BLAST N analysis: *E. coli* O157:H7 str Sakai (EHEC), *E. coli* O157:H7 (EHEC), *E. coli* O157:H7 str F8092B (EHEC), *E. coli* O157 (EHEC), *E. coli* O157:H7 str SS52 (EHEC), *E. coli* O157:H7 str EDL933 (EHEC), *E. coli* O157:H7 str SS17 (EHEC), *E. coli* O157:H- (EHEC), *E. coli* O157:H7 str TW14359 (EHEC), *E. coli* O157:H7 str EC4115 (EHEC), *E. coli* Xuzhou21 (EHEC), *E. coli* O55:H7 (EPEC), *E. coli* O55:H7 str RM12579 (EPEC), *E. coli* O55:H7 str CB9615 (EPEC), *E. coli* O145:H28 (STEC), *E. coli* O145:NM (STEC), *E. coli* O145 str RM9872 (STEC), *E. coli* O145:H28 str RM12581 (STEC), *E. coli* O145:H28 str RM12761 (STEC), *E. coli* O145:H28 str RM13514 (STEC), *E. coli* O145:H28 str RM13516 (STEC), *E. coli* O145 (STEC), *E. coli* O26 str RM10386 (EHEC), *E. coli* O26 str RM8426 (EHEC), *E. coli* O26:H11 str 11368 (EHEC), *E. coli* O26:H11 (EHEC), *E. coli* O39:NM str F8704-2 (ETEC), *E. coli* O45:H2 (STEC), *E. coli* O103 str RM8385 (EHEC), *E. coli* O103:H2 str 12009 (EHEC), *E. coli* O103:H2 (EHEC), *E. coli* O111:NM (STEC), *E. coli* O111:H- str 11128 (STEC), *E. coli* O111:H- (STEC), *E. coli* O111 str RM9322 (STEC), *E. coli* O121 (STEC), *E. coli* O121:H19 (STEC), *E. coli* O121 str RM8352 (STEC), *E. coli* O158:H23 (EPEC), *E. coli* O169:H41 (ETEC), *E. coli* E110019 (EPEC), *E. coli* UMNK88 (ETEC), *Escherichia* sp E4742 (EHEC, Clade II).

The following ExPEC genomes based on unique *E. coli* UTI89 sequence fragments were identified by BLASTN analysis: *E. coli* UTI89 (UPEC), *E. coli* NU14 (UPEC), *E. coli* O25:H1 (UPEC), *E. coli* 536 (UPEC), *E. coli* ABU 83972 (UPEC ABU), *E. coli* UM146 (AIEC/UPEC) and *E. coli* RS218 (NMEC).

Therefore, these fragments can be used to develop multiplex-PCR kits or dot-blot hybridization systems that distinguish both ExPEC and IPEC and also, to some extent, allow differentiation within the pathotype.

TRS-omix extracts all sequences between TRS elements that cover entire genomes (Appendix A). The very restrictive exclusion criteria that we applied allowed us to isolate seven specific DNA fragments (Table 3). One may wonder why only some of the unique sequence regions we selected contained virulence factors typical of the analyzed sub-pathotypes. As we show in Table 4, this is due to the fact that such sequences also show partial similarity to elements of the genomes of other microorganisms.

## 3. Discussion

Many new genomes of various organisms are sequenced every day, generating vast amounts of data. Growing datasets require new, good, fast, easy-to-use, and widely available genome-wide analysis tools. In our paper, a new informatics tool called TRS-omix is presented. It enables genome mapping and extraction of sequences of nucleotides from a genome using specific types of microsatellites known in all genomes, called TRS motifs (included in the flanking sequence). This type of microsatellite was dictated by the fact that they occur more frequently in the studied genomes and are mainly grouped in coding regions [11]. To our knowledge, no software has yet been developed to analyze genomic sequences in this way. The software allows for genomic analysis of linear and circular structures (with a given range or any length of sequence fragments), which offers a wide range of applications and universal use, e.g., prokaryotic and eukaryotic genomes, human chromosomes, etc.

In our research, we have shown that conducting a further study using TRS-omix can lead to the elaboration of a new method of differentiating organisms based on complex systems of TRS motifs. The tool mentioned above can take a fresh look at the genome in the category of a unique arrangement of TRS motifs characteristic of different organisms belonging to an evolutionary group. The approach mentioned above is considered to search for similarities and differences within the sequences contained between TRS motifs for different genomes, which can be significant, while determining specific genetic features of living organisms. In our comparative analysis of two *E. coli* genomes, we have shown that fragments of genomic sequences that are either unique to the ExPEC pathotype or unique to the IPEC pathotype can be found. We also see the potential of using such sequence sets to differentiate genomes/strains within these pathotypes.

We propose strict criteria to distinguish the two selected genomes in our new genetic approach to differentiating IPEC and ExPEC pathotypes (Section Results, Stage 3), as well as the criteria for the elimination of homologous sequences in genomes available in databases and not being sequences belonging only to the group of ExPEC or IPEC pathogens (Section Results, Stage 4). We excluded, among others, a pool of sequences that was specific for the ExPEC pathotype, but also for the AIEC sub-pathotype, which, despite genetic similarity to the UPEC sub-pathotype, nevertheless belongs to the IPEC pathotype [17,18]. This resulted in leaving only three regions characteristic of the IPEC pathotype and two such regions characteristic of the ExPEC pathotype.

For *E. coli* O157: H7 str Sakai genome, the unique regions identified for IPEC, as derived by automated computational analysis using the gene prediction method: the homology of proteins (Vector NTI 11,5), are characterized as follows: the SAK-687 region (1,390,007–1,390,725 bp) partially overlaps the sequence of the CDS 340 gene (hypothetical protein; ncbi: protein/BAB34742.1) and the ureD gene (encoding urease accessory protein D; ncbi: protein/BAB34744.2), the region SAK-937-939 (1,937,664–1,938,325 bp) lies within CDS 626 (encoding hypothetical protein; ncbi: protein/BAB35378.1), and the SAK-1032 region (2,147,832–2,147,944 bp) lies inside the ftrA gene (encoding transcriptional regulator; ncbi: protein/BAB35566.2.

We have not identified any genes that are known markers of virulence for this group of pathogens. As we showed in Table 4, the sequences of these genes are similar to some extent to those of many *E. coli* strains. However, the identified regions are, in our opinion, good markers of the IPEC sub-pathotype since they allowed the identification of 19 genomes of the EHEC sub-pathotype, 16 genomes of the STEC sub-pathotype, five genomes belonging to the EPEC sub-pathotype, and three genomes belonging to the ETEC sub-pathotype.

Similarly, in the case of the *E. coli* UTI89 genome, the unique regions identified for the ExPEC/UPEC genomes, as derived by automated computational analysis using gene the prediction method, Protein homology (Vector NTI 11,5), are characterized as follows: the UTI89-2514 region (4,810,579–4,811,161 bp) partially overlaps the CDS sequence 2305 (encoding hypothetical protein; ncbi: protein/WP_001304629.1) and contains the gene encoding urease accessory protein; ncbi: protein/WP_001335216.1 and the UTI89-2517 region (4,818,430–4,818,864 bp), which partially overlaps the *cnf-1* gene (protein/WP_000528123.1), genes associated with the pathogenicity of the UPEC sub-pathotype.

These sequences allowed the identification of five UPEC genomes (including one genome of ABU strain), one AIEC genome, but with the confirmed high similarity to the uropathogenic UPEC strain (*E. coli* UM146) [19], and one genome belonging to the NMEC sub-pathotype. We also did not identify most of the typical virulence factors of the ExPEC pathotype, which indicates partial similarity of such genomic sequences to elements of genomes of other *E. coli* strains. In this work, we did not conduct this research in more detail. We focused only on using comparative genomic analyses and finding such elements that differentiate the genomes of the two groups of very similar pathogens that can be used as markers of a specific *E. coli* pathotype. Based on the results shown in Table 3, it is possible to design appropriate genetic markers that will enable the differentiation described above by the multiplex-PCR or dot blot hybridization reaction.

The main effect of our interdisciplinary work is, on the one hand, the development of software that allows scientists to automate their research. On the other hand, it enabled the search for new markers differentiating closely related microorganisms. One of the most important functionalities of this software is to support the extraction of sequences between the TRS motifs, determining the position on the genome and some other information necessary for a new genetic approach to the differentiation of IPEC and ExPEC pathotypes. In further studies of the new approach to the deep differentiation of microorganisms, we will try to generalize our method for analyzing entire families of organisms using machine learning or various statistical procedures. This line of research is very promising for potential clinical applications but requires further research in this area from the point of view of biology and computer science.

At the present stage, TRS-omix software is not adapted to the parallel study of thousands of genomes simultaneously, and its scaling is an interesting line of research in this area. In the presented form, TRS-omix allows searching for DNA sequences that may be specific to a particular group of microorganisms; in our case, for the ExPEC or IPEC, two approaches can be used. The first is to subject many genomes from each of these groups to subsequent analyses using TRS-omix and then to check by BLAST N which fragments occur only in one of them and additionally meet the other defined criteria. The second way is to use two genomes for analyses using TRS-omix, one representative for the ExPEC group and one representative for the IPEC group, and then use BLAST N to check which fragments occur only in the genomes of ExPEC strains or only in the genomes of IPEC strains and meet the other defined conditions. In our research, we chose the second approach, as illustrated in Figure 4 and described in Section 2.3. A general approach for differentiation of organisms with the use of TRS-omix.

At present, TRS-omix is a tool supporting the work of researchers, and its mode of operation is sufficient for genetics. Analysis using TRS-omix makes it possible to perform calculations on a personal computer. It seems to us that this is the advantage of TRS-omix because it may be available to many geneticists, for whom this software will enable genome analysis and cheap research in the field of comparing different genomes. We can imagine scalable TRS-omix software computing in big data environments like Apache Hadoop or Apache Spark. However, this approach requires designing the architecture of calculations on individual computing nodes and owning or renting servers with such infrastructure. Using big data technology would enable horizontal scaling (scale-out/scale-in), i.e., adding/removing subsequent machines, and vertical scaling (scale-up/scale-down), i.e., increasing/decreasing resources within one machine.

The TRS-omix is a powerful tool that allows a specific way to compare genomes, particularly those sequences that lie between the trinucleotide repeats. Therefore, it may be an interesting and faster alternative to comparing whole genomes, indicating that these classes of sequences are typical for certain groups of microorganisms.

## 4. Materials and Methods

### 4.1. The Computer Software

The TRS-omix search engine was implemented with the use of a GNU compiler collection called gcc, compliant with ISO 2019 of language C in the programmers’ environment, bearing the name Code::Blocks (release 20.03) under license from GNU. We note that it was developed on the basis of the imperative programming paradigm with the use of the language C, which is dedicated to low level programming [20]. It is worth underlining that the use of language C enables integration of the search engine with other software tools made in popular programming languages such as language R, or Python. In the code architecture of the TRS-omix search engine, data structures corresponding to single-dimensional arrays (linked list) and two-dimensional arrays, which were elaborated based on dynamic memory allocation [21]—the memory was dynamically allocated by the *malloc*() function of language C.

In the case of Linux and Windows operating systems, in order to execute the search of the TRS motifs, it needs inserting into the following in one directory: software file (file name: TRS-omix.exe), defining a file of the class of TRS motifs (file name: *trs.txt*), and a file for analysis (*sequence.fasta*).

In the case of the Linux operating system, we used only the computational server with limited memory access to 8 GB of the Faculty of Mathematics and Computer Science University of Lodz with the following parameters: CPU: 2× AMD EPYC 7302—3 GHz (3.3 GHz Turbo), Cores/Threads: 32C/64T RAM: 128 GB (4× 32 GB) Hard drive: 2× 2.4 TB (Seagate)—RAID1, Operating system: Debian GNU/Linux 10 (buster).

The experimental running tests of the TRS-omix software were carried out on the computer with the following parameters: CPU: Intel Core i7 CPU 860 @ 2.80 GHz, RAM: 8.0 GB RAM, Hard drive: 1 TB., operating system: Windows 10 Professional. In particular, the following genomes were subjected to analysis: *Escherichia coli* UTI89 strain (Acc. no NC_007946.1), a representative of the pathotype ExpEC (Extraintestinal Pathogenic *E. coli*), sub-pathotype UPEC (Uropathogenic/Extraintestinal Pathogenic *E. coli*), and O157:H7 Str. Sakai (Acc. no BA000007.3), a representative of pathotype IPEC (Intestinal Pathogenic *E. coli*), sub-pathotype EHEC (Enterohemorrhagic *E. coli*).

### 4.2. Other IT Software

The Vector NTI 11.5 software using the CLUSTAL W algorithm was used to compare the level of similarity of the sequences extracted from the two investigated genomes by TRS-omix and determine which of these sequences could be pre-qualified unique to a given genome. This software was also used to search for sequences present in a given genome that TRS-omix could not extract because of a mutation within the TRS (imperfect TRS) or because a given complementary sequence contained an internal TRS motif and was therefore assigned to other classes.

The NCBI platform https://blast.ncbi.nlm.nih.gov/Blast.cgi was used to evaluate the available microbial genomes to determine whether a given genomic DNA fragment was specific only for the genome group of the ExPEC or IPEC pathotype. The default settings were used for the analysis, with the exception of the maximum number of hits set to 1000.

### 4.3. Basic Terminology

(1) TRS means a sequence of three nucleotides, e.g., CCG. (2) TRS motif means a sequence of nucleotides, in which there occurs a triple repetition (directly one after another) of the same TRS, e.g., CCGCCGCCG. (3) Class of TRS motifs means TRS motif occurring in one line in the file *trs.txt*, each of which is preceded by the sign „#”, e.g.,: #CCGCCGCCG#CGCCGCCGC#GCCGCCGCC. (4) Number of class of TRS motifs means a natural number (greater than 0), which corresponds to the number of the line in the file *trs.txt*. (5) Flanking sequence means a sequence of nucleotides in which there occurs at least a triple repetition (directly one after another) of the same TRS. (6) Extracted sequence means a sequence of nucleotides (SEQ) that is found between two flanking sequences that consists of at least one nucleotide and is not a flanking sequence. (7) Left flanking sequence (LSF) means a flanking sequence which is found at the site 5′ of an extracted sequence. (8) Right flanking sequence (RSF) means a flanking sequence which is found at the site 3′ of an extracted sequence.4.4. *Using TRS-omix on the example of selected genomes*

The TRS-omix software works with the use of files formed in the FASTA format, which is applied in bioinformatics to record nucleotide sequences representing information on the genome of living organisms and also amino acids in proteins. This offers the possibility of performing analyses with the use of an open-access genome database, e.g., GenBank. Further in this Chapter, we will characterize input files, output files, and software options of the TRS-omix software.

The file called TRS-omix.exe is a workable one of the TRS-omix search engine. It accepts two input files (*sequence.fasta*, *trs.txt*) and creates one output file (*interiors.txt*). The input files should be placed in the same directory as the TRS-omix software. A similar case offers when it comes to the output file—it is formed in the same directory, in which the TRS-omix software finds itself.

The file called *sequence.fasta* contains a genome of the examined organism, with the use of which the TRS-omix search engine searches for microsatellites of the trinucleotide repeats type, and also extracts sequences between such trinucleotide repeat and executes initial analyses of the genome. Let us note that in a file that is downloaded, in an exemplary fashion, from GenBank, it needs merely to change the name of the file into that of “sequence”, but we do not alter the type of file (the type of file is still FASTA). The file *trs.txt* contains classified TRS motifs—each line includes TRS motifs preceded by the sign “#”. One such line is identified as one considered class of TRS motifs. In a similar sense, the file called *trs.txt* is treated as a file with a set of search rules in files of the FASTA type.

The file *interiors.txt* contains information on the positions of flanking sequences and also about those and the very extracted sequences themselves. The first line of the file includes headings of 14 columns, while the following lines contain relevant data. The line including the headlines of the columns was formatted in the following way: L-NoClass;L-No;LFS;Len(LFS);L-POS(LFS);R-POS(LFS);R-NoClass;R-No;RFS;Len(RFS); L-POS(RFS); R-POS(RFS);>SEQ;Len(SEQ) where: L-NoClass—denotes the number of the class of TRS motifs from the file *trs.txt* for the left flanking sequence, L-No—denotes the number of TRS motifs from the file *trs.txt* for the left flanking sequence, LFS—denotes the left flanking sequence; Len(LFS)—denotes the number of nucleotides of the left flanking sequence, L-POS(LFS)—denotes the position from which the left flanking sequence begins in the genome, R-POS(LFS)—denotes the position at which the left flanking sequence ends in the genome, R-NoClass—denotes the number of the class of TRS motifs from the file *trs.txt* for the right flanking sequence, R-No—denotes the number of the TRS motif from the file *trs.txt* for the right flanking sequence, RSF—denotes the right flanking sequence, Len(RFS)—denotes the number of nucleotides of the right flanking sequence, L-POS(RFS)—denotes the position from which right flanking sequence starts in the genome, R-POS(RFS)—denotes the position at which the right flanking sequence ends in the genome, >—denotes the place from which the extracted sequence begins. SEQ—denotes the extracted sequence, Len(SEQ)—denotes the number of nucleotides of the extracted sequence.

Between the elements indicated above there are found semicolons that constitute separators with the omission of a semicolon between the sign “>” and an extracted sequence. Figure 3 illustrates the recording of data in the file *interiors.txt*. The results of the flanking sequence searched in the file *interiors.txt* are recorded in ascending order.

When starting the executable file of TRS-omix, there appear on the computer screen two options which are possible to select:Analysis of the linear case with conditions: use of this option enables to search TRS motifs in linear genomes with additional search conditions: giving the minimal (Min) and maximal (Max) length of the searched sequence found between the flanking sequence.Analysis of the circular case with conditions: use of this option enables searching TRS motifs in circle genomes with analogous additional search conditions like in the case of analysis of the linear case with conditions.

## 5. Conclusions

The results of our studies have shown that it is a new approach to genome-wide analysis using a new bioinformatics tool. Furthermore, it has been discovered that it is possible to use the proposed TRS-omix search engine to find new genetic markers that differentiate IPEC and ExPEC pathotypes. Using this IT tool, we are able to analyze virtually any group of genomes, from viral, through pro- and eukaryotic, to human chromosomes.

The paper presents an efficient search engine for TRS motifs and shows the use of TRS-omix to find new genetic markers according to a novel approach to differentiating pathotypes. The new approach to genome analysis presented in the paper allows one to look at the nucleotide sequences in FASTA files from the point of view of the mapped TRS motif sequences, which enables comparison of the genomes. In addition, the work includes detailed tests of the presented search engine, and also shows an example of an analysis using this bioinformatics tool. The presented informatics tool—TRS-omix—provides a reliable basis for new research on the phylogeny, diagnostics, and epidemiology of organisms. It offers a new look at the genome through the prism of analyses of TRS profiling, which can find application in many areas of biology and medicine in the future.

The tool allows—on the one hand—to carry out analyses using files from open-access bioinformatics databases, such as GenBank, which presently offers a massive amount of data (big data). On the other hand, the tool was created using a low-level language, making its integration with other data science software and tools possible.

## Figures and Tables

**Figure 1 ijms-24-04681-f001:**
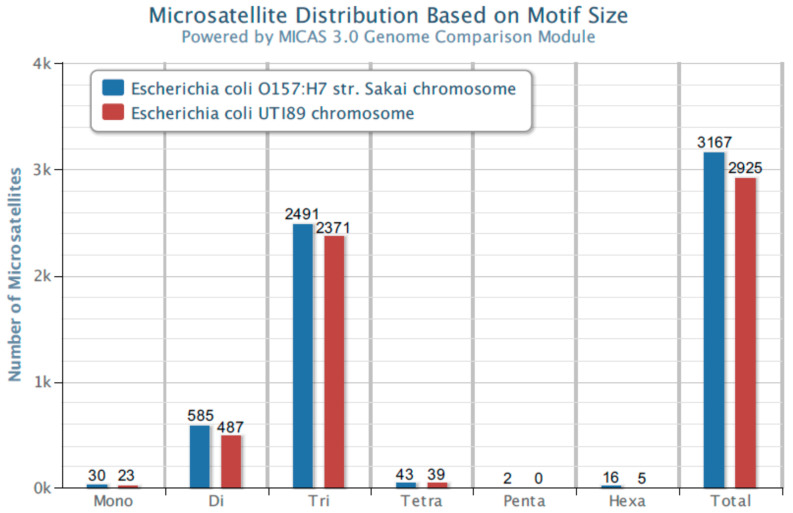
Microsatellite distribution in *E. coli* UTI89 and *E. coli* O157:H7 Sakai. Minimum Iterations: Mono—(9), Di—(4), Tri—(3), Tetra—(3), Penta—(3), Hexa—(3).

**Figure 2 ijms-24-04681-f002:**
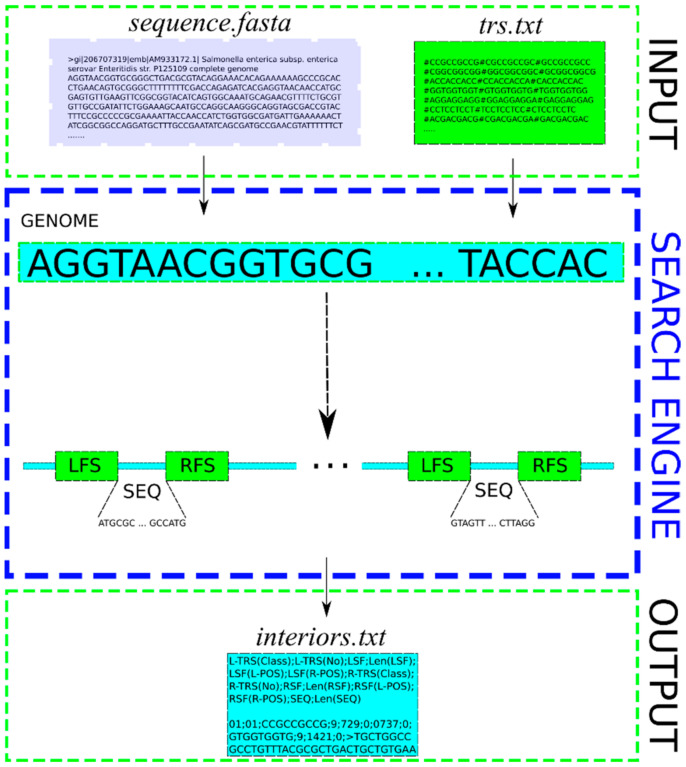
Schematic diagram of the functioning of the engine of TRS-omix. The input data that contain nucleotide sequences in the genome (called *sequence.fasta*) are analyzed for the presence of TRS motifs, and the TRS-omix search engine produces a file (called *interiors.txt*) containing useful information for further genome analysis. Labeling: LFS—means the left flanking sequence (at least triple repetition of the same TRS from the 5′ direction), RFS—means the right flanking sequence (at least triple repetition of the same TRS from the 3′ direction), SEQ—means an extracted DNA sequence (see at https://github.com/TRS-omix/software, accessed on 21 February 2023).

**Figure 3 ijms-24-04681-f003:**
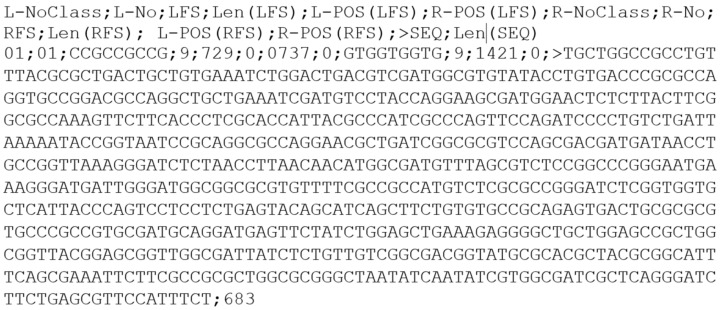
An example of the recording of data for an extracted sequence for *Escherichia coli* strain NCTC9020 (whole genome shotgun sequence). All extracted sequences with information on the positions of flanking sequences are provided in link to GitHub (the file *interiors.txt*): https://github.com/TRS-omix/software/tree/main/Linux_tests/experiement_1 (accessed on 21 February 2023).

**Figure 4 ijms-24-04681-f004:**
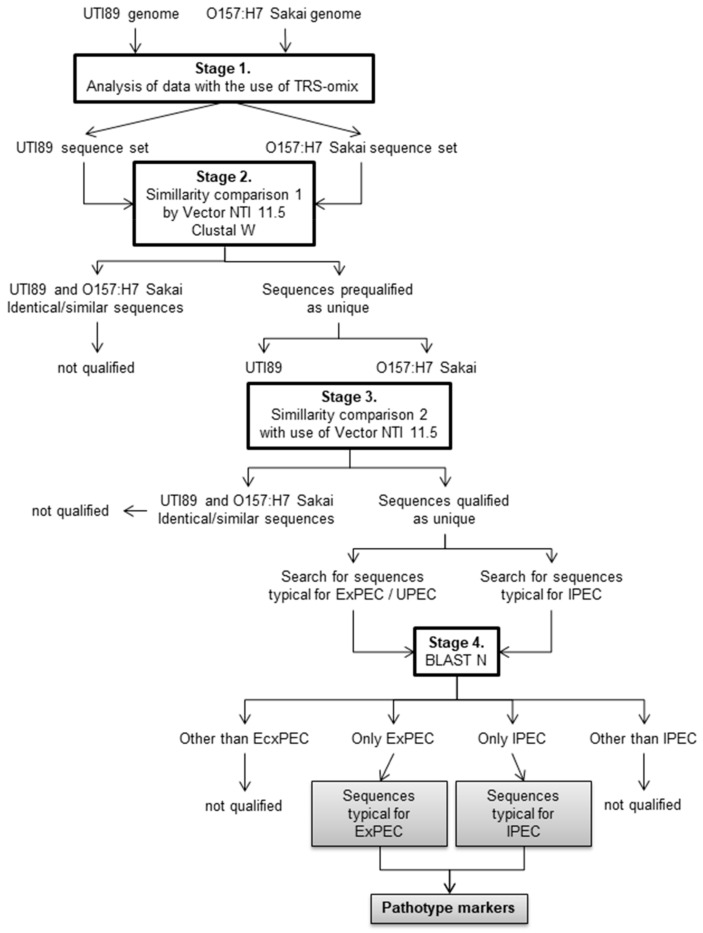
Scheme for analyzing two *E. coli* genomes resulting in filtering out DNA sequences specific only for the ExPEC pathotype or only for the IPEC pathotype (see details in the Results, Analysis of data with the use of TRS-omix and other IT tools section).

**Figure 5 ijms-24-04681-f005:**
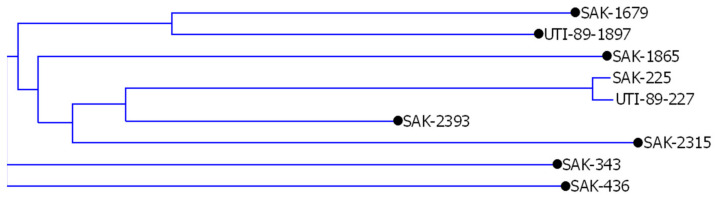
Sequence similarity analysis of the 3′-CCG *n* ≥ 3 and 5′-TCG *n* ≥ 3-flanked sequences extracted from the UTI89 and O157:H7 Sakai genomes. Black dots mark singletons.

**Table 1 ijms-24-04681-t001:** All trinucleotide motifs, excluding mononucleotide repetitions, are grouped in 20 classes. Each class consists of three possible frames of the same motif.

TRS Class No.	TRS Motifs		
1	CCGCCGCCG	CGCCGCCGC	GCCGCCGCC
2	CGGCGGCGG	GGCGGCGGC	GCGGCGGCG
3	ACCACCACC	CCACCACCA	CACCACCAC
4	GGTGGTGGT	GTGGTGGTG	TGGTGGTGG
5	AGGAGGAGG	GGAGGAGGA	GAGGAGGAG
6	CCTCCTCCT	TCCTCCTCC	CTCCTCCTC
7	ACGACGACG	CGACGACGA	GACGACGAC
8	CGTCGTCGT	GTCGTCGTC	TCGTCGTCG
9	AGCAGCAGC	GCAGCAGCA	CAGCAGCAG
10	GCTGCTGCT	CTGCTGCTG	TGCTGCTGC
11	AACAACAAC	ACAACAACA	CAACAACAA
12	GTTGTTGTT	TTGTTGTTG	TGTTGTTGT
13	AAGAAGAAG	AGAAGAAGA	GAAGAAGAA
14	CTTCTTCTT	TTCTTCTTC	TCTTCTTCT
15	TACTACTAC	ACTACTACT	CTACTACTA
16	GTAGTAGTA	TAGTAGTAG	AGTAGTAGT
17	ATCATCATC	TCATCATCA	CATCATCAT
18	GATGATGAT	ATGATGATG	TGATGATGA
19	ATAATAATA	TAATAATAA	AATAATAAT
20	TATTATTAT	ATTATTATT	TTATTATTA

**Table 2 ijms-24-04681-t002:** Experimental results obtained on the computational server (Min = 200, Max = 3000).

No.	Description (from GenBank)	GenBank (Information)	Input Size (bp) (*sequence.fasta*)	Output Size (Bytes) (*interiors.txt*)	Time (s) (Execution)	Memory (%) (Execution)
1	*Escherichia coli* NCTC9020 (whole genome)	NZ_UFYE01000002.1 (linear case)	4,617,408 bp	2,164,354 bytes	3.709 s	2.7%
2	*Human chromosome* 14 (complete sequence)	AL954800.2 (linear case)	87,191,216 bp	37,404,928 bytes	69.147 s	12.3%
3	*Salmonella enterica* B71 (complete sequence)	KP899806.1 (circular case)	190,730 bp	46,026 bytes	0.163 s	1.2%
4	*Salmonella enterica* LT2 (complete genome)	AE006468.2 (circular case)	4,857,450 bp	2,613,817 bytes	4.132 s	5.6%

**Table 3 ijms-24-04681-t003:** The list of genomic fragments of *E. coli* UTI89 identified exclusively in the ExPEC *E. coli* genomes, and genomic fragments of *E. coli* O157:H7 Sakai identified exclusively in the IPEC *E. coli* genomes (black dots).

			IPEC GENOMES																																											ExPEC GENOMES						
			EHEC	EHEC	EHEC	EHEC	EHEC	EHEC	EHEC	EHEC	EHEC	EHEC	EHEC	EPEC	EPEC	EPEC	STEC	STEC	STEC	STEC	STEC	STEC	STEC	STEC	EHEC	EHEC	EHEC	EHEC	ETEC	STEC	EHEC	EHEC	EHEC	STEC	STEC	STEC	STEC	STEC	STEC	STEC	EPEC	ETEC	EPEC	ETEC	EHEC, CLADE II	UPEC	UPEC	UPEC	UPEC	UPEC (ABU)	AIEC/UPEC	NMEC
**FRG**	**FRG [BP]**	**REGION**	*E. coli* O157:H7 str Sakai	*E. coli* O157:H7	*E. coli* O157:H7 str F8092B	*E. coli* O157	*E. coli* O157:H7 str SS52	*E. coli* O157:H7 str EDL933	*E. coli* O157:H7 str SS17	*E. coli* O157:H-	*E. coli* O157:H7 str TW14359	*E. coli* O157:H7 str EC4115	*E. coli* Xuzhou21	*E. coli* O55:H7	*E. coli* O55:H7 str RM12579	*E. coli* O55:H7 str CB9615	*E. coli* O145:H28	*E. coli* O145:NM	*E. coli* O145 str RM9872	*E. coli* O145:H28 str RM12581	*E. coli* O145:H28 str RM12761	*E. coli* O145:H28 str RM13514	*E. coli* O145:H28 str RM13516	*E. coli* O145	*E. coli* O26 str RM10386	*E. coli* O26 str RM8426	*E. coli* O26:H11 str 11368	*E. coli* O26:H11	*E. coli* O39:NM str F8704-2	*E. coli* O45:H2	*E. coli* O103 str RM8385	*E. coli* O103:H2 str 12009	*E. coli* O103:H2	*E. coli* O111:NM	*E. coli* O111:H- str 11128	*E. coli* O111:H-	*E. coli* O111 str RM9322	*E. coli* O121	*E. coli* O121:H19	*E. coli* O121 str RM8352	*E. coli* O158:H23	*E. coli* O169:H41	*E. coli* E110019	*E. coli* UMNK88	Escherichia sp E4742	*E. coli* UTI89	*E. coli* NU14	*E. coli* O25:H1	*E. coli* 536	*E. coli* ABU 83972	*E. coli* UM146	*E. coli* RS218
SAK-687	719	S1	●	●	●	●	●	●	●		●	●	●				●	●	●	●	●	●	●	●	●	●	●	●	●	●	●	●	●	●	●	●	●	●	●		●		●	●								
SAK-937	187	S2	●	●		●																														●	●			●												
SAK-938	123		●	●		●		●																												●	●															
SAK-939	334		●	●		●		●																												●	●															
SAK-1032	113	S3	●	●	●	●	●	●	●	●	●	●	●	●	●	●																										●			●							
UTI89-2514	583	U1																																												●	●	●	●	●	●	●
UTI89-2517	435	U2																																												●	●	●		●	●	●

**Table 4 ijms-24-04681-t004:** Selected genomic fragments of the O157:H7 Sakai and UTI89 genomes found within sequences of typical IPEC/ExPEC virulence factors present in bacterial genomes.

VF	SEQUENCE NO	GENOMES
*stx1B*	SAK-1396	IPEC, SHIGELLA, E. ALBERTII
*stx1B*, *stx1A*	SAK-1397	IPEC, SHIGELLA, ENTEROBACTER CLOACAE, AEROMONAS CAVIAE
*stx1A*	SAK-1398	IPEC, SHIGELLA, ENTEROBACTER CLOACAE, AEROMONAS CAVIAE
stx2A. stx2B	SAK-642	IPEC, ACINETOBACTER HAEMOLYTICUS, ENTEROBACTER CLOACAE
*eae*	SAK-2331	IPEC, E. ALBERTII, SHIGELLA, CITROBACTER
*efa1*	SAK-1906	IPEC, E.ALBERTII, LOW SCORE OTHERS
*efa1*	SAK-1907	IPEC, VARIOUS SPECIES
*cnf1*	UTI-89-2516	EXPEC, IPEC
*cnf1*	UTI-89-2518	EXPEC, IPEC
*sfaA*, *sfaD*, *sfaE*, *sfaF*, *sfaG*	UTI-89-563	EXPEC, IPEC
*hlyA*	UTI-89-2519	EXPEC, IPEC, SHIGELLA
*hlyA*	UTI-89-2520	EXPEC, IPEC, SHIGELLA
*hlyA*	UTI-89-2521	EXPEC, IPEC, SHIGELLA
*hlyA*, *hlyC*	UTI-89-2522	EXPEC, IPEC, SHIGELLA

## Data Availability

Data are available on request from the corresponding author. All data for the performed computational experiments are on the GitHub: https://github.com/TRS-omix/software (accessed on 21 February 2023).

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
