# Peer review of "New Genetic Markers Differentiating IPEC and ExPEC Pathotypes—A New Approach to Genome-Wide Analysis Using a New Bioinformatics Tool"

_ijms, 2023, doi:10.3390/ijms24054681_

Round 1
Reviewer 1 Report (Previous Reviewer 1)
I believe the manuscript has been greatly improved. The advanced answers are satisfactory.Author Response
Please see the attachment

Reviewer 2 Report (New Reviewer)
Dear authors,
here is my central comment:
- this is a central point to me. Is this tool scalable to thousands of genomes? Because to identify markers that robustly discern between genotypes/pathotypes/lineages, you need a scalable tool and a robust machine learning and/or statistical approach to measure accuracy, reproducibility, sensitivity, and specificity using training and validating datasets while considering the population structure of the species.
The authors mention using computer with multiple cores for testing purposes, but looks like the code is not written to support multiple cores, so there is no benefit from that. How hard would it be to make the code parallel and scalable, considering the size of bioinformatics data?
But here are other comments to be considered as well:
- Some sentences don't use proper English syntax, or are incomplete (e.g., lines 32, 91, 109, 193, 332, 409).
- The authors sometimes use BLAST N, and sometimes BLAST throughout the paper, so please make this uniform.
- The developers should provide compilation and usage instructions in their repo, as well as more explanatory README file and detailed description of the format of input and output files.
- One of the main bottlenecks of bioinformatics tools is their RAM memory consumption, which is not compared in this paper. I would suggest the authors to include this information in their computational comparisons as well.
- The current implementation of TRS-omix assumes that the input files are named sequence.fasta and all the files are in the same directory as the executable. This may not be very practical when working on high-performance computers and multiple fasta files, since all input files need to be named the same.
- The developers provide different options for circular and linear genomes, so what is the difference in the TRS-omix implementation between the two?
- How does TRS-omix perform for genomes with multiple chromosomes?
- I find the naming of the output columns a bit confusing and hard to understand without checking their meaning.
- How does the data structure used in TRS-omix perform compared to using k-mers to extract the flanking sequences?
I hope this is helpful to improve the quality of the work.
Thank you
Author Response
Please see the attachment

This manuscript is a resubmission of an earlier submission. The following is a list of the peer review reports and author responses from that submission.
Round 1
Reviewer 1 Report
Bioinformatics is a field of scientific activity at the interface of biology and computer science. It has developed numerous algorithms and programs intended to analyze, visualize and manipulate three-dimensional molecular structures related to genomics. In this sense, the present work has focused essentially on a bacterial model (in particular the genomes of Escherichia coli) for the design of a potential bioinformatics software (TRS-omix) to genome-wide analysis. The authors suggest that this tool could be considered as a search engine for detecting useful genetic markers for the differentiation of various patotypes. All the results obtained show that such a tool could be considered as an interesting alternative for comparing genomes and detecting typical sequences at a fairly rapid rate. Overall, the document is quite well written and documented. It reflects a certain consistency in the presentation of the results obtained. It is considerable work. It reflects that bioinformatics resources could have a tangible role in the development of research in several fields, including biology and medicine.Nevertheless, here are some comments:
- In line 381, please change “2” to “4”
- In line 382, please change “2.1” to “4.1”
- In line 411, please change “2.2” to “4.2”
- In line 423, please change “2.3” to “4.3”
- In line 439, please change “2.3” to “4.4”.
- Is this bioinformatics software scalable?
- A very brief conclusion !
Reviewer 2 Report
This bioinformatics tool proposed by Majchrzak, et al. utilizes an array of 20 classes of trinucleotide repeats and their 3 potential frame shifts distributed throughout the E. coli genome to create regions for comparion. The number of Microsattalites based upon trinuceotides for UPec and IPEC are similar, 2371-2491 microsatelites. The advantage of this TRS-omics software, is the ability to extract and analyze the sequences between the TRS motifs which could facilitate evolutionary comparisons of the E. coli genomes. Unfortunately, I feel that the 20 classes of R-TRS may be distributed throughout the genome of both E. coli at reasonably high frequencies, but the ability of the resulting portions of the genome to contain comparable or unique sequences is limited. The fact that known virulence genes that are present in O157:H7 (SLT, eae, etc,) and known to be absent from ExPEC were not identified by this method is a concern. Similarly, there may be hemolysin genes, cytonecrotizing factor, and other virulence genes unique to ExPEC that were not identified. Only two genes of interest, ureD and ftrA were identified, and only 6 total unique sequences were identified. Location or number of TRS, the size limitation of the intervening sequences (>100 bp) or the required similarity >90%) may be preventing identification of additional or unknown unique virulence genes. From the perspective of a diagnostic microbiologist or someone studying bacterial pathogenesis, this is a limitation. I feel that inclusion of Supplemental Excel Files S4 and S5 or just pertinent regions of these data would be helpful to the reader in following the interiors.txt analysis and the resulting specificity listing. Perhaps they could be included in the body of manuscript.

Round 2
Reviewer 2 Report
I still do not feel it is appropriate to exclude pathogen genes such as slt, eae,cnf, and many others, when comparing the IPEC to ExPEC just because such genes may be found in other genera or species of Enterobacteriaceae. This would bias the interpretation of unique sequences versus conserved genes and their proportion in two strains being compared, potentially masking linkages that may have had an evolutionary role.